# Fibular notch morphometry and its clinical importance on dry bones

**Nazire Kilic Safak** [ORCID] *

Department of Anatomy, Cukurova University Faculty of Medicine, Adana, Turkey

* nazirekilic84@gmail.com

## Abstract

The aim of the present study was to determine the morphometric characteristics of the fibular notch (FN). This study was carried out with 76 dry adult tibial bone specimens (right 38, left 38) with unknown age and sex collected from the Department of Anatomy, Cukurova University, Adana. The mean width of the FN was 23.04 ± 2.02 mm; the mean depth of the FN 3.63 ± 0.83 mm; the mean height of the FN was 41.76 ± 4.01 mm. The mean anterior facet length and posterior facet length was found to be 10.44 ± 1.94 mm and 13.93 ±1.63 mm, respectively. The mean value of the angle between the anterior and posterior facets was found to be 140.56˚ ± 11.72. The mean value of the angle between the anterior surface of the tibia and the intertubercular line was 75.5˚ ± 5.47. No statistically significant differences were detected between the right and left sides for all measurements. It is considered that knowing the morphometric and anatomical characteristics of the fibular notch in detail will help radiologists evaluate the talocrural region. It is also considered that these data will guide surgeons and help determine the appropriate size for ankle reconstruction operations.

**Data Availability Statement:** All information about this data can be obtained via the corresponding author.

**Funding:** The author(s) received no specific funding for this work.

## Introduction

The interosseous edge of the tibia is sharp and forms an attachment point for the interosseous membrane, which connects the leg bones. This sharp edge turns into a groove called "fibular notch" toward the lower end of the tibia. The fibular notch (FN) provides a fibrous attachment area where the distal end of the fibula is located [1]. For the ankle joint to function properly, this relationship between the distal end of the tibia and fibula is extremely important [2]. Ankle joint dislocations and fractures are quite common. Knowing the syndesmosis joint structure here is important in terms of treatment and prognosis. However, obtaining radiological findings on injuries in the area is very difficult because of rotational differences, shape differences, and variations [3]. The ankle joint is among the most injured joints [4]. Although the movement of the tibiofibular joint is quite limited, it has a significant effect on stabilizing ankle movements. Fibular notch morphological features are associated with displacement in the distal tibiofibular syndesmosis and ankle sprains [5]. In cases of ankle injuries, it is important to place the fibula correctly at the fibular notch to maintain the stability of the talus at the tibiofibular joint or ankle mortise [6]. It is very important to have a comprehensive knowledge of the morphometric characteristics of the fibular notch to stabilize the joint, make an accurate

**Competing interests:** The authors have declared that no competing interests exist.

diagnosis, and ensure success in operations that may be performed in the area [2]. Additonally, fibular notch approach is a safe and reliable method preferred for surgeries of the ankle. It is extremely important to know the anatomy of the region to determine the feasibility of this approach and comprehensive anatomical studies are still needed in literature [7]. Fibular notch measurements are important for assessing most appropriate position of the distal fibula in fibular notch for ankle stability. In accordance with this purpose, the aim of the present study was to determine the morphometric characteristics of the fibular notch.

## Materials and methods

This study was carried out with 76 dry adult tibial bone specimens (right 38, left 38) with unknown age and sex collected from the Department of Anatomy, Cukurova University, Adana. All necessary permits were obtained for the described study, which complied with all relevant regulations. The ethical approval was obtained from the Cukurova University Faculty of Medicine Non-Interventional Clinical Research Institutional Ethics Committee (2023, 133). Samples were accessed in August and September 2023 for research purposes. Damaged and deformed bones were excluded from this study. All measurements were taken twice by the same researcher and mean values were recorded (NKS).

Images were taken with Canon 50D Digital Camera from a distance of 70 cm after illuminating with artificial light by using a reference scale of fixed length (10 mm). The measurements were made by using ImageJ Software (https://imagej.net/ij/) after the images were transferred to the computer. This program uses real world measurement units such as millimeters.

The following parameters were measured:

1. **Fibular Notch Width (FNW):** The distance between the anterior and posterior tubercles of the fibular notch is measured from the widest part.

2. **Fibular Notch Depth (FND):** It is measured from the deepest point, perpendicularly to a line between the anterior and posterior tubercles of the fibular notch.

3. **Fibular Notch Height (FNH):** The distance between the highest point of the tibial plafond and the highest point of the fibular notch is measured.

4. **Anterior Facet Length (AFL):** The distance between the tuberculum anterior of the tibia and the deepest point of the fibular notch is measured.

5. **Posterior Facet Length (PFL):** The distance between the tuberculum posterior of the tibia and the deepest point of the fibular notch is measured.

6. **Angle between Anterior and Posterior Facet (AAPF):** The angle between the anterior and posterior facet is measured.

7. **Angle between the Anterior Surface of the Tibia and the Intertubercular Line (AASIL):** The angle between the line that passes tangent to the anterior surface of the tibia and the intertubercular line is measured (Figs 1 and 2).

The data were analyzed with the IBM SPSS V20. Compliance with normal distribution was examined with the Shapiro-Wilk Test. When right and left side measurements were compared, the comparisons were made with the independent samples $t$-test because the data followed a normal distribution. The Pearson Correlation Analysis was used to evaluate the relationship between the measurement values and the significance level was taken as $p < 0.05$.

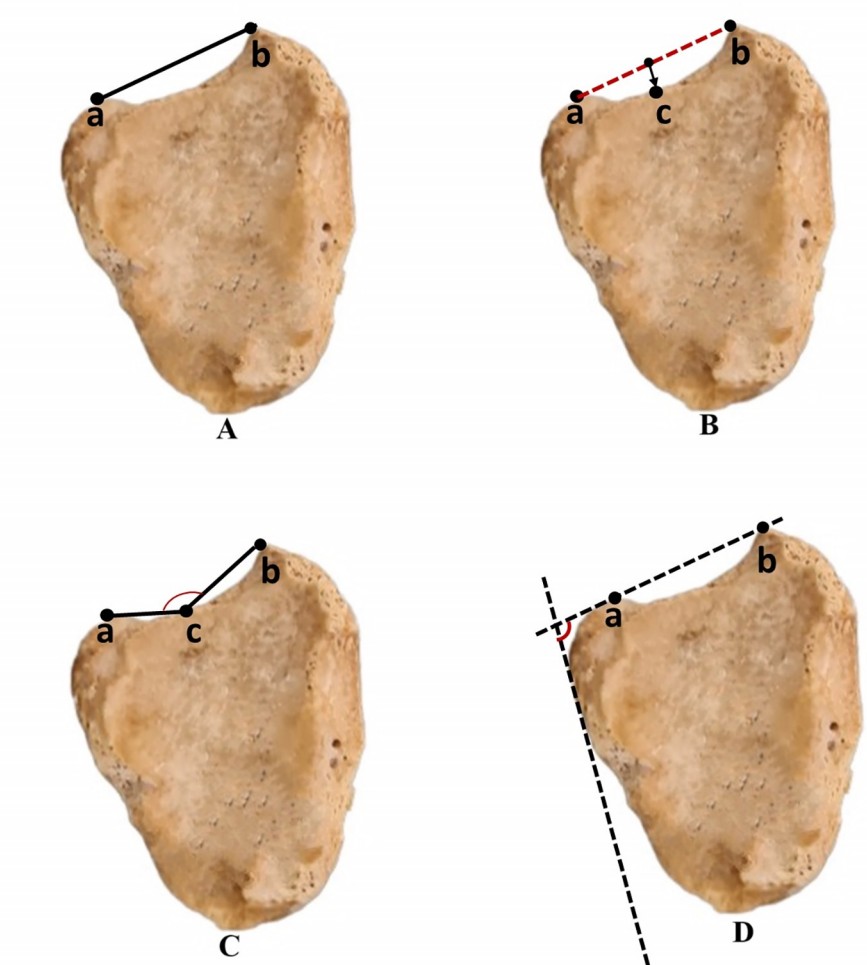

**Fig 1.** A) FNW: from a to b; B) FND: Perpendicular distance from c to intertubercular line (a-b); C) AFL: From a to c; PFL: from b to c; AAPF: Angle between a-c and b-c; D) AASIL: Angle between intertubercular line (a-b) and anterior surface tangent line.

## Results

In the present study, the mean fibular notch width was found to be 23.04 ± 2.02 mm. The mean value was obtained as 22.91 ± 1.79 mm for the right side and 23.18 ± 2.25 mm for the left side. No significant differences were detected between the mean values of fibular notch width in terms of fibular notch width (p = 0.575).

Fibular notch depth was found to be 3.61 ± 0.77 mm, 3.59 ± 0.72 mm, and 3.63 ± 0.83 mm in total, on the right side, and left side, respectively, and no significant differences were detected between the two sides (p = 0.836).

In this study, the mean fibular notch height was found to be 41.76 ± 4.01 mm in total (42.19 ± 3.83 mm for the right side and 41.32 ± 4.19 mm for the left side. No statistically significant differences were detected between the right and left sides in terms of fibular notch height measurements (p = 0.349).

The anterior facet length was found to be 10.43 ± 2 mm on the right side, 10.46 ± 1.91 mm on the left side, and 10.44 ± 1.94 mm in total, and the mean values did not differ in terms of

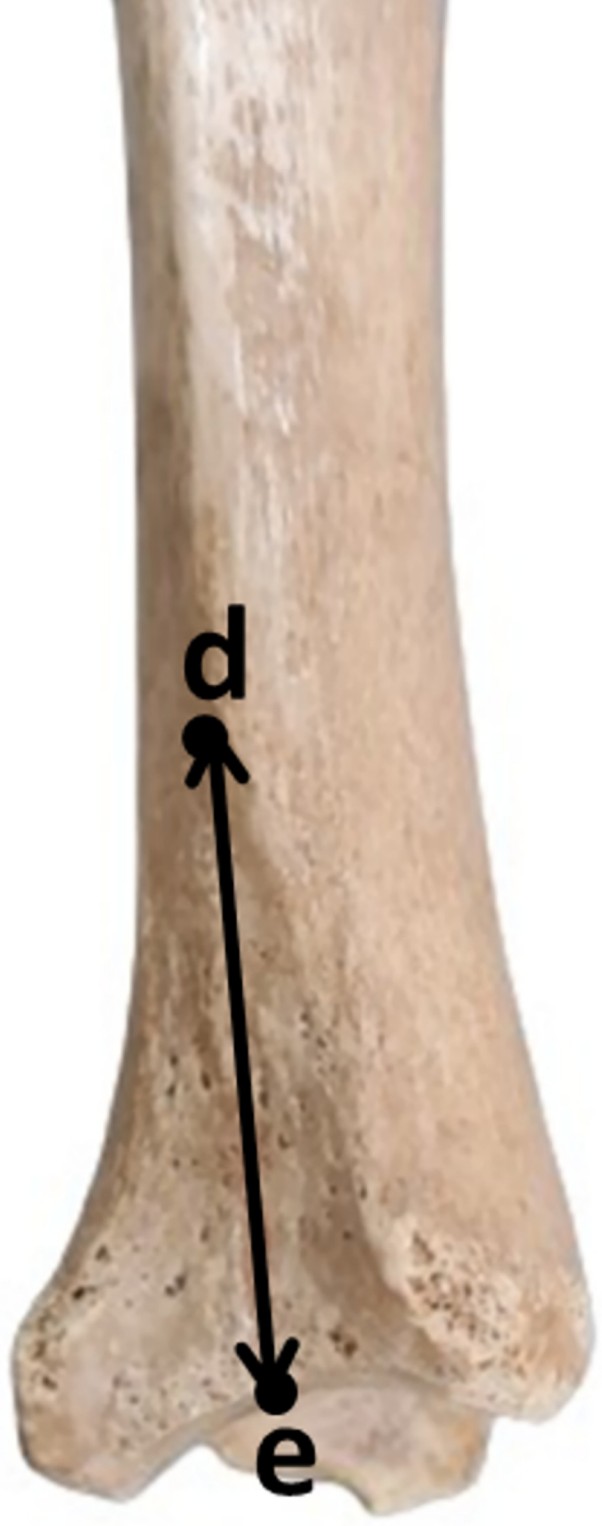

**Fig 2. Fibular notch height measurement (FNH: from d to e).**

**Table 1. Measurement of the fibular notch.**

|  | Right (n = 38) | Left (n = 38) | Total (n = 76) | p* |
|---|---|---|---|---|
| FNW (mm) | 22.91 ± 1.79 | 23.18 ± 2.25 | 23.04 ± 2.02 | 0.575 |
| FND (mm) | 3.59 ± 0.72 | 3.63 ± 0.83 | 3.61 ± 0.77 | 0.836 |
| FNH (mm) | 42.19 ± 3.83 | 41.32 ± 4.19 | 41.76 ± 4.01 | 0.349 |
| AFL (mm) | 10.43 ± 2 | 10.46 ± 1.91 | 10.44 ± 1.94 | 0.946 |
| PFL (mm) | 13.71 ± 1.85 | 14.14 ± 1.52 | 13.93 ± 1.69 | 0.271 |
| AAPF (˚) | 141.37 ± 12.66 | 139.75 ± 10.79 | 140.56 ± 11.72 | 0.550 |
| AASIL (˚) | 76.03 ± 5.27 | 74.98 ± 5.68 | 75.5 ± 5.47 | 0.408 |

*Independent sample *t*-test; Mean ± standard deviation

(FNW: fibular notch width, FND: fibular notch depth, FNH: fibular notch height, AFL: anterior facet length, PFL: posterior facet length, AAPF: angle between anterior and posterior facet, AASIL: angle between the anterior surface of the tibia and the intertubercular line)

sides (p = 0.946). The posterior facet length was found to be 13.71 ± 1.85 mm for the right side, 14.14 ± 1.52 mm for the left side, and a total of 13.93 ± 1.69 mm. No significant differences were detected between the right and left sides in terms of posterior facet length (p = 0.271).

The mean value of the angle between the anterior and posterior facets was found to be 141.37˚ ± 12.66, 139.75˚ ± 10.79, and 140.56˚ ± 11.72 on the right, left and in total, respectively. The angle between the anterior and posterior facets did not have a significant difference between the right and left sides in terms of mean values (p = 0.55).

The mean value of the angle between the anterior surface of the tibia and the intertubercular line was 76.03˚ ± 5.27. The mean values of the right and left sides were 74.98˚ ± 5.68 and 75.5˚ ± 5.47, respectively, and no statistically significant differences were detected between the two sides (p = 0.408).

An overview of the measurements and side-comparisons in Table 1.

A positive and statistically significant relationship was detected between fibular notch width and anterior and posterior facet length (*r* = 0.232 and *r* = 0.428, respectively). There was a significant and negative relationship between fibular notch depth and the angle between the anterior and posterior facets (*r* = -0.380). A positive and statistically significant relationship was detected between the fibular notch height and posterior facet length (r = 0.270). Likewise, a positive and statistically significant relationship was detected between the posterior facet length and the angle between the anterior and posterior facet (r = 0.230). No statistically significant relationships were detected between other measurements (Table 2).

**Table 2. Correlation analysis results between measurements.**

|  | 1 FNW | 2 FND | 3 FNL | 4 AFL | 5 PFL | 6 AAPF | 7 AASIL |
|---|---|---|---|---|---|---|---|
| 1-FNW | — |  |  |  |  |  |  |
| 2-FND | 0.016 |  |  |  |  |  |  |
| 3-FNL | 0.046 | -0.028 |  |  |  |  |  |
| 4-AFL | **0.232** | 0.184 | 0.013 |  |  |  |  |
| 5-PFL | **0.428** | 0.023 | **0.270** | 0.066 |  |  |  |
| 6-AAPF | 0.159 | **-0.380** | -0.063 | 0.147 | **0.230** |  |  |
| 7-AASIL | 0.121 | -0.109 | -0.138 | -0.201 | -0.022 | 0.133 | — |

*Pearson correlation analysis

(FNW: fibular notch width, FND: fibular notch depth, FNH: fibular notch height, AFL: anterior facet length, PFL: posterior facet length, AAPF: angle between anterior and posterior facet, AASIL: angle between the anterior surface of the tibia and the intertubercular line)

## Discussion

The tibiofibular syndesmosis joint is very important for the normal functioning of the ankle [3]. This joint, which is also called the distal tibiofibular joint, is formed between the convex surface on the medial of the fibula and the triangular-shaped fibular notch located on the lateral surface of the distal tibia, creating a settlement for the trochlea of the talus [8]. A total of 13% of ankle fractures are accompanied by tibiofibular syndesmosis injury. The diagnosis of tibiofibular syndesmosis injury is very difficult because of shape variations of the tibial tubercles and anatomical variations of fibular notch depth [9]. Having comprehensive knowledge about distal tibiofibular syndesmosis is very important in the pre- and post-operative evaluations of ankle sprains and fractures [10]. The fibular notch approach is a reliable, successful, and innovative approach in complex distal tibia fractures. More anatomical studies are required to determine the feasibility of this approach [7]. There are measurements in the literature on CT or MRI images, cadavers, and plastination slices [10–12]. The fact that the fibular notch is located more posteriorly with a lower angle is associated with the anterior talofibular ligament injuries [13]. In their study conducted a study on MRI images, Yıldırım et al. argued that all measurements except anterior-posterior facet angle differed according to gender and were higher in men than women [10].

Mavi et al. reported that the fibular notch depth was deeper in individuals with recurrent ankle sprains [14]. Fibular notch depth was reported to be 4.1 ± 1.8 mm in men, 4.1 ± 1.6 mm in women, and 4.1 ± 1.7 mm in the entire population in a study that was conducted by Tonogai et al. on CT images [15]. Yıldırım et al. reported that the fibular notch depth was 3.6 mm for men and 2.9 mm for women in the measurements made on the images obtained with the MRI method [10]. In a study conducted on CT images of cadavers, fibular depth was found to be 4.29 ± 1.26 mm [11]. In their CT-based study, Misir et al. reported that the depth was 3.3 ± 0.9 mm, 3.3 ± 0.9 mm, and 3.0 ± 0.9 mm in the entire population, men and women, respectively [16]. In a study that was conducted on CT images in Asians, the depth was found to be 4.1 ± 0.9 mm on the right and 4.4 ± 1.0 mm on the left, and no differences were detected between two sides [17]. In their study conducted on 20 plastinated male cadavers, Sora et al. reported the mean fibular depth as 5.07 ± 0.76 mm [12]. In their study conducted on dry bones, Musa et al. reported that the fibular notch depth was 3.40 ± 0.96 mm in men and 2.9 ± 0.66 mm in women [18]. In a study conducted on 100 dry tibias in India, it was reported to be 2.85 ± 1.38 mm for the right side and 2.90 ± 1.57 mm for the left side [19]. Also, the results of the study conducted on dry bones in India showed the depth as 1.08 cm and no significant differences were detected between the right and left facets [3]. Shivaji et al. reported that the fibular notch depth was 3.25±032 mm for men and 3±0.25 for women in dry bones [20]. In a study conducted with multi-detector Computed Tomography in China, depth measurements were reported to be 5.1 mm in men and 4.2 mm in women in 3-D images, 5.0 mm in men, and 4.3 mm in women in 2-D axial images [9]. In a study that was conducted on dry bone samples in the Czech Republic, the mean fibular notch depth was reported to be 4.5 ± 1.2 mm [21]. In a retrospective study conducted on CT images of 60 healthy adults in Athens, the fibular notch depth was indicated as 3.92 mm [22]. Taşer et al. reported it as 3.68 ± 1.49 mm in the morphometric evaluations they made on 35 dry adult tibial bone specimens [2]. The fibular depth measurement was found to be similar to the results of Taşer et al in our study. The mean result of the fibular notch depth in the present study was smaller than Czech and Asian population, gerater than Indian and Kenyan population.

Chen et al. reported that the fibular notch height was 35.1 mm in male and 33.7 mm in female in their study conducted with Multi-Detector Computed Tomography [9]. In a study that was conducted on CT images in the Turkish population, fibular notch height was found

to be 27.2 ± 2.9 mm in the entire population, 27.8 ± 2.9 mm in men, and 25.4 ± 2.3 mm in women [16]. In a study that was conducted in India by using the direct measurement method on dry bones, the fibular notch height was found to be 38.82 ± 5.58 mm for the right side and 38.72 ± 7.68 mm for the left side [19]. Shivaji et al. reported the height as 31.87 ± 2.01 mm and 29.32 ± 1.5 mm for men and women, respectively [20]. In a study that was conducted by direct measurement on dry bones, the fibular notch height was found to be 31.41 ± 3.66 mm in men and 29.85 ± 2.33 mm in women, and a difference was detected between genders [18]. In measurements that were performed on dry bones in Turkey, fibular notch height was reported to be 29.43 ± 4.07 mm [2]. In a study conducted in the Czech Republic, the mean fibular notch height was reported to be 42.5 ± 5.6 mm in dry bones [21]. When we compare our study result with other studies, the mean value of the fibular notch height in the present study was smaller than Czech population and greater than Indian, Kenyan and Chinese population.

In a previous study that was conducted on the CT images in Korea, the fibular notch width was reported to be 23.8 ± 2.0 mm for the right side and 23.2 ± 2.1 mm for the left side, and no statistically significant differences were detected between two sides [17]. In another study conducted on a cadaver in Austria, it was found that the fibular notch width was 23.76 ± 2.57 mm [12]. The measurements made on dry bones in Kenya revealed that the fibular notch width was 22.2 ± 1.83 mm and 20.52 ± 1.83 mm in men and women, respectively. It was reported in the same study that a significant difference was detected between men and women in terms of fibular notch width [18]. Fojtik et al. reported that the mean fibular notch width was 23.6 ± 2.6 mm in dry bones [21]. In another study that was conducted in India, the fibular notch width was reported to be 2.33 cm, and no differences were detected between the right and left sides [3]. In another study conducted in India, the fibular notch width was found to be 23.94 ± 1.02 mm in men and 20.91 ± 1.14 mm in women [20]. Sreedevi et al. reported that the fibular notch width in dry bones was 17.42 ± 2.29 mm and 16.83 ± 2.01 mm for right and left, respectively [19]. In their study conducted on dry bones, Taşer et al. showed that the fibular notch width was 23.26 ± 3.11 mm [2]. Our study results are similar to the studies of Taşer et al. in terms of fibular notch width. The mean value of the fibular notch width in the present study was smaller than Czech, Korean and Austrian population and greater than Kenyan and Indian population.

In a study focused on relationship between ankle instability and fibular notch morphometric measurements, demonstrated that there was a strong correlation between ankle instability and anterior facet length. A presence of narrow anterior facet indicated that lateral ankle instability. In this study anterior facet length were found as 11.3 mm in study group and 13.0 mm in control group [23]. In a previous study that was conducted by using the MRI results, anterior and posterior facet lengths were reported to be 10.4 mm in men and 8.9 mm in women [10]. Musa et al. reported the anterior facet length in dry bones as 11.49 ± 1.97 mm in men and 10.9 ± 1.60 mm in women. They reported in the same study that the posterior facet length was 16.84 ± 2.13 mm and 16.08 ± 1.42 mm in men and women, respectively [18]. The anterior facet length was reported to be 11.63 ± 0.78 mm in men and 9.11 ± 0.69 mm in women in India, and the posterior facet length was 16.37 ± 0.99 mm in men and 11.65 ± 1.06 mm in women [20]. In the measurements that were made with a digital caliper on dry bones, anterior and posterior facet lengths were 1.34 and 1.28 cm, respectively. It was also reported in the study that no differences were detected between the two sides [3]. The mean anterior facet length value was reported to be 13.19 ± 1.96 mm and 12.6 ± 1.49 mm for the right and left facets, respectively. Posterior facet length was reported as 15.71 ± 2.03 mm for the right facet and 15.66 ± 1.41 mm for the left facet. In the same study, no differences were detected between the facets [19]. Taşer et al. reported that the anterior facet length was 10.89 ± 2.08 mm and the posterior facet length was 13.28 ± 1.49 mm [2]. The mean value of the anterior and posterior facet

length in our study was smaller than Kenyan and Indian population and similar with the study results of Taşer et al.

In the measurements made with a goniometer on dry bones in India, the angle between the anterior and posterior facets was reported to be 124.6° ± 6.34 for the right side and 126.1° ± 8.24 for the left side [19]. In the study conducted by Yıldırım et al., the angle between the anterior and posterior facets was reported to be 138.6° and 139.9° in men and women, respectively [10]. The mean value of the the angle between the anterior and posterior facets in our study was greater than Indian population and similar with studies in our region which conducted by Yıldırım et al.

The angle between the anterior surface of the tibia and the intertubercular line, which is considered to affect the superposition of the tubercles in anteroposterior radiography images, was measured as 74.63° ± 3.13 on average in the study conducted by Taşer et al [2]. In our study, the mean value was obtained as 75.5° ± 5.47, which is similar to this result.

## Conclusion

It is considered that knowing the morphometric and anatomical characteristics of the fibular notch in detail will help radiologists evaluate the talocrural region. It is also considered that these data will guide surgeons and help determine the appropriate size for ankle reconstruction operations.

## Supporting information

**S1 File.**
(XLSX)

## Acknowledgments

The author present thanks to Resul Safak for his contribution.

## Author Contributions

**Conceptualization:** Nazire Kilic Safak.

**Data curation:** Nazire Kilic Safak.

**Formal analysis:** Nazire Kilic Safak.

**Investigation:** Nazire Kilic Safak.

**Methodology:** Nazire Kilic Safak.

**Project administration:** Nazire Kilic Safak.

**Supervision:** Nazire Kilic Safak.

**Writing – original draft:** Nazire Kilic Safak.

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
