## [Decision Letter · Decision Letter 0]

8 Apr 2024

PONE-D-23-34676Fibular Notch Morphometry and Its Clinical Importance on Dry BonesPLOS ONE

Dear Dr. KILIÇ ŞAFAK,

Thank you for submitting your manuscript to PLOS ONE. After careful consideration, we feel that it has merit but does not fully meet PLOS ONE’s publication criteria as it currently stands. Therefore, we invite you to submit a revised version of the manuscript that addresses the points raised during the review process.

We look forward to receiving your revised manuscript.

Kind regards,

Piotr Karauda, Ph.D.

Academic Editor

PLOS ONE

Journal Requirements:

3. In your manuscript, please provide additional information regarding the specimens used in your study. Ensure that you have reported human remain specimen numbers and complete repository information, including institution name and geographic location. 

For more information on PLOS ONE's requirements for paleontology and archeology research, see https://journals.plos.org/plosone/s/submission-guidelines#loc-paleontology-and-archaeology-research.

4. In the online submission form, you indicated that all information about this data can be obtained via the corresponding author.

Reviewers' comments:

Reviewer's Responses to Questions

**Comments to the Author**

1. Is the manuscript technically sound, and do the data support the conclusions?

Reviewer #1: Yes

Reviewer #2: Yes

2. Has the statistical analysis been performed appropriately and rigorously? 

Reviewer #1: Yes

Reviewer #2: Yes

3. Have the authors made all data underlying the findings in their manuscript fully available?

Reviewer #1: Yes

Reviewer #2: Yes

4. Is the manuscript presented in an intelligible fashion and written in standard English?

Reviewer #1: Yes

Reviewer #2: Yes

5. Review Comments to the Author

Reviewer #1: 1. In discussion the authors have mentioned what other got but not comparted their results with his results.

2. In figure 1 all measurements are not seen like angles etc. (AFD: from a to c; PFD: from b to c; AAPF: angle between a-c and bc; AASIL: angle between intertubercular line and anterior surface tangent line.) Show this correctly.

Kindly do the corrections as mentioned above. Kindly add some new clinically relevant information to your study.

Reviewer #2: 1. The photos of tibia can be included with labeling and measurements taken instead of adding hand drawn images

2. Below the table-1 the expansion (full form) of various parameters can be given for easy interpretation

3. The Pearson Correlation Analysis was used to evaluate the relationship between the measurement values but the analysis is not clearly explained in the result and interpreted like t test.

4. The references were till 2020 if possible the latest reference can be given

5. out of 76 dry bones how many were right and how many were left, Sex identification of the bone was done or not explain

6. ImageJ Software include the software details and the precision point of the software

7. Aim & objectives can be explained clearly with the interpretations

6. PLOS authors have the option to publish the peer review history of their article (what does this mean?). If published, this will include your full peer review and any attached files.

Reviewer #1: **Yes: **Dr. Chandni Gupta

Reviewer #2: No

---

## [Author Response · Author response to Decision Letter 0]

24 May 2024

Responses for Journal Requirements:

1. The requested checks completed and amandments about PLOS ONE's style requirement have been made.

2. All information about this data uploaded as supplementary information tool. 

3. An additional information regarding the specimens used in your study added in our manuscript. 

4. All information about this data uploaded as supplementary information tool. 

Responses for Reviewer 1:

1. In accordance with the reviewer’ suggestion, authors were added comparisons with other studies in discussion part.

2. All measurements in Figure 1 have been corrected. New clinically relevant information were added to our study.

Responses for Reviewer 2:

1. The photos of tibia added instead of hand drawn images in line with the recommendations.

2. Below the Table1 the full form of various parameters is given for easy interpretation.

3. Additional table (Table 2) has been added for clearly understanding of analysis. 

4. The relevant latest references have been added. 

5. As mentioned in the first sentence of materials and methods section “This study was carried out with 76 dry adult tibial bone specimens (right 38, left 38) with unknown age and sex collected from the Department of Anatomy, Cukurova University, Adana.”.

6. Detailed information about the ImageJ Software and precision point were added in the manuscript. And also for additional information web address of the software were added in the manuscript. 

7. Aim & objectives were updated for explaining clearly with the interpretations.

I would like to thank the editors and reviewers for their valuable contributions.

---

## [Decision Letter · Decision Letter 1]

1 Jul 2024

Fibular Notch Morphometry and Its Clinical Importance on Dry Bones

PONE-D-23-34676R1

Dear Dr. KILIÇ ŞAFAK,

We’re pleased to inform you that your manuscript has been judged scientifically suitable for publication and will be formally accepted for publication once it meets all outstanding technical requirements.

Kind regards,

Piotr Karauda, Ph.D.

Academic Editor

PLOS ONE

Additional Editor Comments (optional):

Congratulations, well done!

Reviewers' comments:

Reviewer's Responses to Questions

**Comments to the Author**

1. If the authors have adequately addressed your comments raised in a previous round of review and you feel that this manuscript is now acceptable for publication, you may indicate that here to bypass the “Comments to the Author” section, enter your conflict of interest statement in the “Confidential to Editor” section, and submit your "Accept" recommendation.

Reviewer #1: All comments have been addressed

Reviewer #2: All comments have been addressed

2. Is the manuscript technically sound, and do the data support the conclusions?

Reviewer #1: Yes

Reviewer #2: Yes

3. Has the statistical analysis been performed appropriately and rigorously? 

Reviewer #1: Yes

Reviewer #2: Yes

4. Have the authors made all data underlying the findings in their manuscript fully available?

Reviewer #1: Yes

Reviewer #2: Yes

5. Is the manuscript presented in an intelligible fashion and written in standard English?

Reviewer #1: Yes

Reviewer #2: Yes

6. Review Comments to the Author

Reviewer #1: Thanks for doing all the corrections. Well done the study has been done meticulously and presented well.

Reviewer #2: (No Response)

7. PLOS authors have the option to publish the peer review history of their article (what does this mean?). If published, this will include your full peer review and any attached files.

Reviewer #1: **Yes: **Dr. Chandni Gupta, Additional Professor, Department of Anatomy, KMC, Manipal

Reviewer #2: No

---

## [Editor Report · Acceptance letter]

5 Jul 2024

PONE-D-23-34676R1 

PLOS ONE

Dear Dr. KILIÇ ŞAFAK, 

I'm pleased to inform you that your manuscript has been deemed suitable for publication in PLOS ONE. Congratulations! Your manuscript is now being handed over to our production team.

Kind regards, 

on behalf of

Dr. Piotr Karauda 

Academic Editor

PLOS ONE